# Orthogonalization speed-up from quantum coherence after a sudden quench

**Beatrice Donelli[1,2]★, Gabriele De Chiara[3,4], Francesco Scazza[5,6] and Stefano Gherardini[1,2]†**

**1** Istituto Nazionale di Ottica del Consiglio Nazionale delle Ricerche (CNR-INO), 50125 Firenze, Italy.
**2** European Laboratory for Non-linear Spectroscopy, Università di Firenze, 50019 Sesto Fiorentino, Italy.
**3** Física Teòrica: Informació i Fenòmens Quàntics, Departament de Física, Universitat Autònoma de Barcelona, 08193 Bellaterra, Spain.
**4** Centre for Quantum Materials and Technology, School of Mathematics and Physics, Queen's University Belfast, Belfast BT7 1NN, United Kingdom.
**5** Department of Physics, University of Trieste, 34127 Trieste, Italy.
**6** Istituto Nazionale di Ottica del Consiglio Nazionale delle Ricerche (CNR-INO), 34149 Trieste, Italy.

★ beatrice.donelli@ino.cnr.it

## Abstract

We introduce a nonequilibrium phenomenon, reminiscent of Anderson's orthogonality catastrophe (OC), that arises in the transient dynamics following an interaction quench between a quantum system and a localized defect. Even if the system comprises only a single particle, the overlap between the asymptotic and initial superposition states vanishes according to a power-law scaling with the number of energy eigenstates entering the initial state and an exponent that depends on the interaction strength. The presence of quantum coherence in the initial state is reflected onto the discrete counterpart of an infinite discontinuity in the system spectral function, a hallmark of Anderson's OC, as well as in the quasiprobability distribution of work due to the quench transformation. The positivity loss of the work distribution is directly linked with a reduction of the minimal time imposed by quantum mechanics for the state to orthogonalize. We propose an experimental test of coherence-enhanced orthogonalization dynamics based on Ramsey interferometry of a trapped cold-atom system.

# 1   Introduction

Characterizing the dynamical response of quantum systems to time-dependent perturbations is a central aspect of nonequilibrium quantum physics [1]. A key feature of such dynamics is the incompatibility between the initial state and the observables of interest [2–7]. In particular, for a quantum system subjected to a time-dependent (external) drive, the initial state typically does not commute with the instantaneous Hamiltonian at later times. A widely studied case is a quench transformation of the system Hamiltonian, where the initial state does not commute with the post-quench Hamiltonian [8–12]. Less explored is a more extreme case, which we will consider here, in which the initial state is not even an eigenstate of the initial Hamiltonian [13,14]. This stronger kind of incompatibility [15–17] can radically affect both the dynamic and thermodynamic response of a quantum system subjected to perturbations. Recent applications of such scenarios include quantum phase estimation [18], work extraction beyond classical limits [19], thermalization towards a non-Abelian thermal state [20], and the generation of nonequilibrium steady states in the framework of coherent collision models [21,22].

Over the past decade, experimental advances have enabled unprecedented control for manipulating quantum states and accessing quantum coherent regimes, allowing studies of nonequilibrium dynamics and thermodynamics. In this regard, platforms such as nitrogen-vacancy centers in diamond [23–25], nuclear magnetic resonance setups [26–28], trapped ions [29], neutral atoms [30–33], and optomechanical systems [34] have proven to be particularly successful. However, to our knowledge, dynamical behaviours such as decay scaling directly attributed to the incompatibility of the initial state with the system Hamiltonian before and after the applied quench transformation have not yet been fully characterized. The application of such dynamical effects, leading to potential quantum advantage in perturbation sensing and information processing, e.g. quantum state engineering, remains elusive.

In this paper, we investigate a system of quantum particles confined in a one-dimensional harmonic potential and suddenly coupled to a spatially localized defect. Such a defect could be realized in ultracold atomic mixtures by the combination of state- or species-selective localizing potentials with contact interactions [30, 35–38]. Within a zero-range approximation valid in the low-energy limit, atomic interactions can be effectively modelled through a delta-like Fermi pseudopotential. In particular, in the center-of-mass reference frame, the problem of two interacting atoms reduces to an effective single-particle dynamics subject to a delta-like perturbation when the potential is purely harmonic [39].

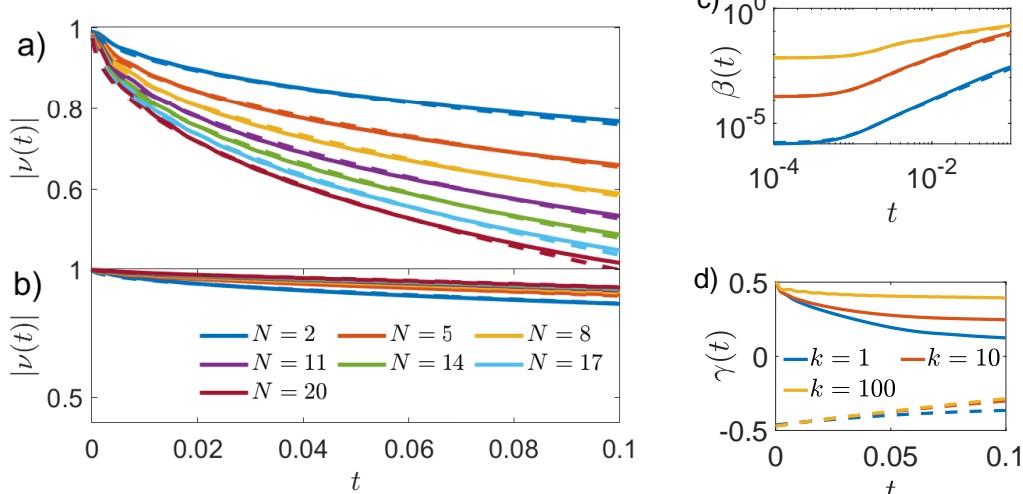

Figure 1: Decay of the LE for a single particle initialized in the state of Eq. (3), after the interaction with a delta perturbation of strength $k$ is switched on. Panels (a)-(b): Time-behaviour of $|\nu(t)|$ taking the superposition state of Eq. (3) [panel (a)] or the corresponding diagonal state of Eq. (4) [panel (b)] as initial states, for $k = 100$ and $N \in \{2, 5, 8, 11, 14, 17, 20\}$. The solid lines represent the values of $|\nu(t)|$ obtained numerically, while the dashed lines are the fitted curves following the scaling law of Eq. (1). Panels (c)-(d): Time dependence of $\beta(t), \gamma(t)$ for $k = 1, 10, 100$; here solid lines refer to taking $|\psi(0)\rangle$ equal to the superposition state, while the dashed lines are associated to the corresponding diagonal state.

Under the effect of a delta-like perturbation, we demonstrate the existence of genuine quantum regimes where state-orthogonalization emerges over time, even at the single-particle level or in a fermionic gas made of few particles. Notably, we uncover striking similarities between the post-quench dynamics of a quantum system initialized in a pure superposition state, and the so-called orthogonality catastrophe (OC) [38, 40, 41] of a Fermi sea initialized in its ground state and perturbed by a local scatterer. To unveil this coherence-induced orthogonalization, we analyze the Loschmidt echo (LE) [6, 9, 42–44], which is connected to dynamical quantum phase transitions [45–49], and investigate work statistics through quasiprobabilities [14, 19, 50, 51]. In particular, we find the following scaling law for the decay of the modulus of the LE $|\nu(t)|$:

$$|\nu(t)| \sim 1 - \beta(t)N^{\gamma(t)}, \tag{1}$$

where $\beta(t)$ and $\gamma(t)$ are time-dependent coefficients, and $N$ is the number of Hamiltonian eigenstates in the initial superposition state. The key quantity in Eq. (1) is the decay exponent $\gamma$ that is positive for initial states that include quantum coherences and negative otherwise, as shown in Fig. 1. Moreover, positive $\gamma$ tend to saturate with increasing defect strength. A positive $\gamma$ results in a much faster decay of $|\nu(t)|$ as $N$ increases, which leads to an *orthogonalization speed-up*. A direct manifestation of this effect is the discrete counterpart of an infinite discontinuity in the quasiprobability distribution of work done by the defect, whose real part coincides with the spectral function of the system. The spectral function is the main quantity considered by Nozières and De Dominicis to develop the dynamical theory of the OC [52]. The infinite discontinuity and subsequent power-law decay of the distribution of work is compatible with the discontinuity present in the single-particle spectrum of a perturbed Fermi sea undergoing Anderson's OC [38, 41].

| $k$ | Superposition case | | Diagonal case | |
|---|---|---|---|---|
| | $\beta(t)$ | $\gamma(t)$ | $\beta(t)$ | $\gamma(t)$ |
| $10^2$ | $0.6\,t^{0.52}$ | $0.36\,t^{-0.042}$ | $0.48\,t^{0.47}$ | $-0.46+1.82\,t$ |
| $10$ | $0.85\,t^{0.98}$ | $0.15\,t^{-0.21}$ | $0.73\,t^{1.01}$ | $-0.46+1.63\,t$ |
| $1$ | $0.06\,t^{1.34}$ | $0.044\,t^{-0.49}$ | $0.06\,t^{1.36}$ | $-0.45+0.95\,t$ |

Table 1: Fitted values $\beta(t)$ and $\gamma(t)$ of Fig. 1, initializing a single particle in the superposition state of Eq. (3) or in the corresponding diagonal state of Eq. (4), for $k = 1, 10, 100$.

## 2 Decay scaling

Coherence-enhanced orthogonalization takes place during the transient dynamics following the quench transformation that operates the interaction with a localized defect. Such an orthogonalization is driven by quantum coherence in the initial state with respect to the pre- and post-quench Hamiltonian. The delta-function potential, modelling the interaction with the localized defect at $x = 0$, is $\hat{V} = k\,\delta(\hat{x})$ where the defect strength $k$ is restricted to positive values. This perturbation leads to the quenched Hamiltonian $\hat{H}' = \hat{H} + k\,\delta(\hat{x})$ where $\hat{H}$ is the initial Hamiltonian of the system. The system sensitivity to the perturbation is quantified by the LE

$$\nu(t) = \text{Tr}\left[\hat{U}'(t)\,\hat{\rho}(0)\hat{U}(-t)\right], \qquad (2)$$

where $\hat{\rho}(0)$ is the initial state and $\hat{U}(t) = e^{-i\hat{H}t}$ ($\hat{U}'(t) = e^{-i\hat{H}'t}$) is the evolution operator due to $\hat{H}$ ($\hat{H}'$). The LE can be measured using a Ramsey interferometric scheme [6,41,53].

### 2.1 Single particle in an initial equal superposition state

Let us consider a single particle in an harmonic trap, initialized in a pure state $|\psi(0)\rangle$. The initial Hamiltonian $\hat{H}$ of the particle is the one of the one-dimensional quantum harmonic oscillator: $\hat{H} = -\frac{1}{2}\frac{d^2}{dx^2} + \frac{1}{2}\omega^2\hat{x}^2$ with frequency $\omega$. As a first example, we take $|\psi(0)\rangle$ as an equal superposition of the $N$ lowest-energy eigenstates of the unperturbed Hamiltonian $\hat{H}$:

$$|\psi(0)\rangle = \frac{1}{\sqrt{N}}\sum_{\substack{n=0\\n\text{ even}}}^{N}(-1)^{n/2}|n\rangle, \qquad (3)$$

where $n$ is an even integer number and $|n\rangle$ is the $n$-th eigenstate of $\hat{H}$. We consider $n$ even because the eigenstates of $\hat{H}$ with $n$ odd are not affected by the delta perturbation.

The scaling of $|\nu(t)|$ yields information about the dynamical properties of the system affected by the defect. Panels (a)-(b) of Fig. 1 show the decay of $|\nu(t)|$ as a function of time $t$. To do that, we take as initial states both $\hat{\rho}(0) = |\psi(0)\rangle\langle\psi(0)|$, outer product of the pure superposition state $|\psi(0)\rangle$ of Eq. (3), and the corresponding mixed diagonal state

$$\hat{\rho}_{\text{diag}}(0) = \frac{1}{N}\sum_{n=0,\text{even}}^{N}|n\rangle\langle n| \qquad (4)$$

that does not contain quantum coherence terms along the basis of the unperturbed Hamiltonian. We fit the decay of the system using Eq. (1), which depends on $t$ and $N$.

The time-behaviour of the multiplicative factor $\beta(t)$ in (1) is independent on the presence of quantum coherence in the initial state along the basis of the unperturbed Hamiltonian; see panel (c). Thus, the marked difference of the curves in panels (a)-(b) lies in the sign and behaviour of

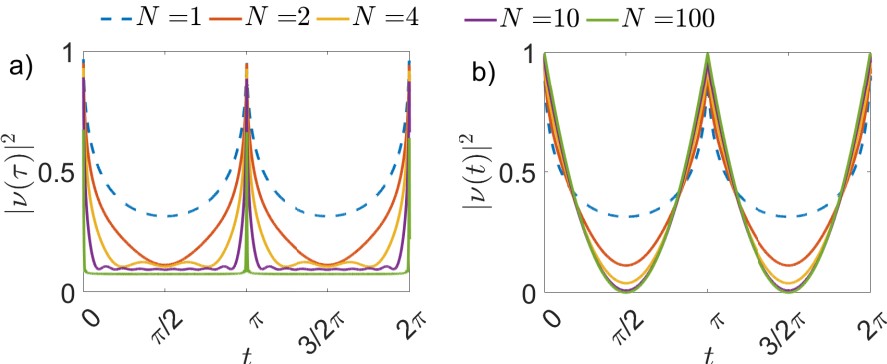

Figure 2: Squared modulus of LE as a function of time, initializing the system in the superposition state of Eq. (3) [panel (a)] and in the corresponding diagonal state of Eq. (4) [panel (b)], with $k = 1000$ for $N = 1, 2, 4, 10, 100$. All the lines are solid, except for the dashed one that represents the reference case $N = 1$ where the initial state is the ground state.

$\gamma(t)$, whose fits are in panel (d). When the initial state has quantum coherence, $\gamma(t)$ is positive for any $t$. This induces a drastic change in the state of the system, which tends to an almost orthogonal state with respect to the initial condition. The corresponding orthogonalization time decreases as $1/N$ at maximum, as we will show later using arguments based on the quantum speed limit. Increasing the number $N$ of energy eigenstates in $|\psi(0)\rangle$ leads to a speed-up of the orthogonalization process. On the contrary, if the quantum system is initialized in a diagonal state, i.e., an incoherent mixture, increasing $N$ slows down the orthogonalization. The intensity $k$ of the delta perturbation does not alter significantly the trends of the initial decay of $|\nu(t)|$ in Fig. 1. This fact is evident in panels (c)-(d) of Fig. 1 and Table 1, showing respectively the trends and expressions of $\beta(t)$ and $\gamma(t)$ for $k = 1, 10, 100$. Interestingly, $\gamma$ tends to become constant for $k$ large (in Fig. 1, $\gamma \approx 0.41$ for $k = 100$).

In Fig. 2 we illustrate the time dependence of $|\nu(t)|^2$ for the single particle, in the limit of strong intensity $k \to +\infty$ of the delta perturbation. Notice that we have plotted $|\nu(t)|^2$ instead of $|\nu(t)|$ for resolution purposes. In both panels of the figure, $|\nu(t)|^2$ exhibits a periodic behaviour with period $T = \pi\omega^{-1}$ for any $N$. This trend is characterized by periodic cusps that indicate the presence of quantum recurrences occurring at times when the quantum system returns to its initial configuration. In Appendix A we report a formal derivation of the cuspid behaviour of $|\nu(t)|^2$ for a relevant case-study. The period $T$ separating two cusp singularities can be explained by noting that the minimum energy gap of the system is $\Delta E_{\min} = 2\hbar\omega$, which is due to the symmetry of its states. Therefore, the oscillation main period of $|\nu(t)|^2$ is $T = \frac{2\pi}{\min(\Delta E)} = \pi$, as we numerically observed.

As $N$ increases, the system remains at the minimum of $|\nu(t)|$ for a longer duration if quantum coherence are included in the initial state, as shown in panel (a), thus highlighting a more stable orthogonalization process. On the contrary, without the contribution of coherences in the initial state, the LE does not flatten with increasing $N$ [panel (b)] but still remains peaked at $t = j\pi$, with $j$ integer, exhibiting a change in the initial decay curvatures.

The presence of periodic cusp singularities in $|\nu(t)|$ is an hallmark of a dynamical phase transition (DPT). As discussed in Refs. [47, 49] and references therein, non-analyticities in the LE usually manifest as cusps in its time behaviour. In our case, the periodicity of the cusps aligns with the characteristic time scale dictated by the energy-gap structure of the perturbed system. This phenomenon has a universal trait in the sense that it generally manifests in a system that undergoes a sudden perturbation, whereby a sharp revival structure in the LE signals critical time-behaviours. It is worth noting that, differently to the canonical manifestation of DPT phenomenology in many-

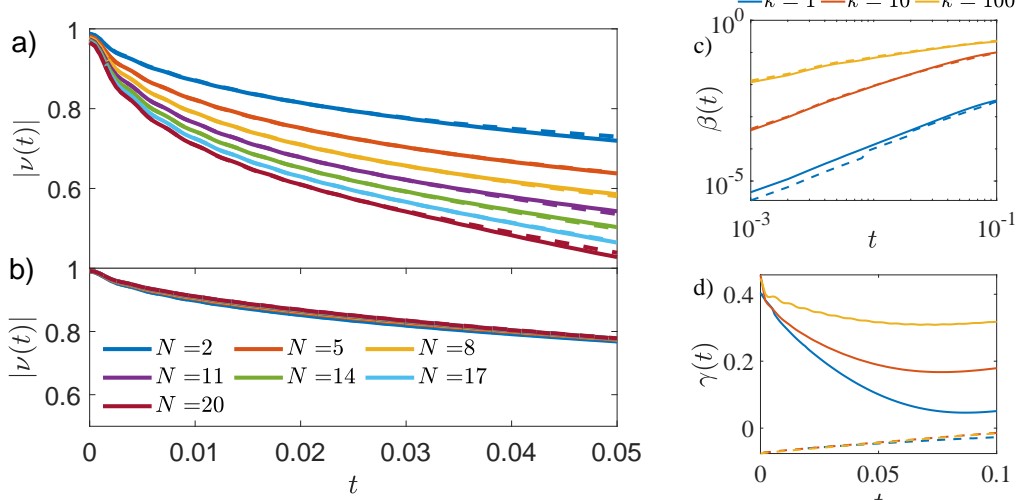

Figure 3: Time evolution the LE modulus for two fermions subjected to a delta perturbation. (a)-(b) $|\nu(t)|$ as a function of time, for two fermions initialized in the coherent antisymmetrized superposition of energy eigenstates (5) [panel (a)] and in the corresponding incoherent diagonal state [panel (b)], respectively. Here, the fermions are perturbed by a delta-function potential of strength $k = 100$. Solid lines represent the numerical results, while dashed lines refer to the fitted curves according to the scaling law of Eq. (1). Panels (c)-(d): time-dependence of $\beta(t)$ and $\gamma(t)$ for different perturbation strengths $k$, both in the coherent (solid lines) and incoherent case (dashed lines).

body system, the single-particle case lacks a spatial order parameter that prevents the revival of $|\nu(t)|$ to 1.

## 2.2 Analysis with two fermions

In this section we report the effect of a localized defect, modelled as a delta perturbation, on two fermions that are initialized in the anti-symmetric state

$$|\psi(0)\rangle = \frac{1}{\sqrt{2N}} \sum_{n=2}^{N} (-1)^{n/2} \Big( |0\,n\rangle - |n\,0\rangle \Big). \tag{5}$$

In Fig. 3, we illustrate the time evolution of $|\nu(t)|$ and fit it using the scaling law $|\nu(t)| \sim 1 - \beta(t)M^{\gamma(t)}$ of Eq. (1). Panel (a) refers to the initial anti-symmetric state of Eq. (5), while panel (b) corresponds to the associated diagonal state where all the off-diagonal terms are removed. Solid lines represent the numerical results, while dashed lines correspond to the fitted curves following the scaling law (1). In table 2 we report the fitted coefficients $\beta(t)$ and $\gamma(t)$, whose time-dependence is displayed in panels (c)-(d) of Fig. 3, for $k = 1, 10, 100$. As in the other figures, the coherent (solid lines) and incoherent (dashed lines) scenarios are compared. While $\beta(t)$ remains nearly unchanged between the two cases, $\gamma(t)$ exhibits a sign change, becoming positive when quantum coherences are present. The decay rate increases with the superposition size $M$, while it decreases when off-diagonal elements of the initial density operator are accounted for. This behaviour highlights a coherence-enhanced orthogonalization effect, whereby quantum coherence accelerates the decay of $|\nu(t)|$ with increasing $M$. These results mirror the scaling law observed in the single-particle case, as the LE exhibits a quite similar power-law scaling. Importantly, however, here the scaling is shaped by the anti-symmetrization of the two fermions affected by the delta perturbation.

| $k$ | Superposition case | | Diagonal case | |
|---|---|---|---|---|
| | $\beta(t)$ | $\gamma(t)$ | $\beta(t)$ | $\gamma(t)$ |
| $10^2$ | $0.70\,t^{0.49}$ | $0.25\,t^{-0.080}$ | $0.74\,t^{0.49}$ | $-0.07+0.60\,t$ |
| $10$ | $0.88\,t^{0.92}$ | $0.09\,t^{-0.25}$ | $0.84\,t^{0.93}$ | $-0.07+0.59\,t$ |
| $1$ | $0.056\,t^{1.32}$ | $0.003\,t^{-1.12}$ | $0.06\,t^{1.32}$ | $-0.07+0.41\,t$ |

Table 2: Fitted values $\beta(t)$ and $\gamma(t)$ of Fig. 5, initializing two fermions in the superposition state of Eq. (5) or in the corresponding diagonal state, for $k = 1, 10, 100$.

In Fig. 4, we show that, as $N$ increases, the function $|\nu(t)|^2$ becomes progressively steeper, thus inducing the quantum speed-up of the orthogonalization process. Again, initializing the system in a diagonal state, without quantum coherence along the Hamiltonians bases, prevents these effects.

While Anderson's OC refers to a scaling with the number of fermions, our analysis focuses on the scaling of $|\nu(t)|$ with the number $N$ of states in the initial superposition. This leads to the intriguing interpretation that the decay of the system is somewhat reminiscent of that experienced by a many-particles Fermi sea in Anderson's OC.

## 3   Role of quantum coherence & physical interpretation

The Fourier transform of the LE $\nu(t)$ returns the Kirkwood-Dirac quasiprobability (KDQ) distribution [6, 13, 14, 19, 51, 54] $P(w) = \sum_{n,m} q_{n,m}\delta(w - w_{n,m})$ of the work done by the defect on the quantum system. In this formula, $w_{n,m} = E'_m - E_n$ is the energy difference between the final perturbed eigenvalue $E'_m$ and the initial unperturbed one $E_n$. The relation between the LE and the KDQ distribution of work is thus $\nu(t) = \sum_{n,m} q_{n,m}e^{-iw_{m,n}t}$. Considering a generic initial superposition state $|\psi(0)\rangle = \sum_n \alpha_n |\psi_n\rangle$, the KDQs assume the expression

$$q_{n,m} = \alpha_n^* \Lambda_{m,n}^* \sum_k \Lambda_{m,k}\alpha_k, \tag{6}$$

where $\Lambda_{m,n} \equiv \langle \psi'_m|\psi_n\rangle$ is the overlap between the $m$-th eigenstate of the perturbed Hamiltonian $\hat{H}'$ and the $n$-th eigenstate of the unperturbed one $\hat{H}$. In the limit of $k \to +\infty$, the overlaps $\Lambda_{m,n}$ can

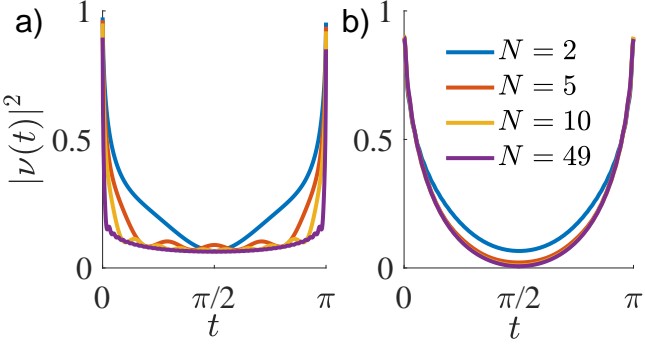

Figure 4: Time-behaviour of $|\nu(t)|^2$ for two fermions initialized in the anti-symmetric state of Eq. (5) [panel (a)] and in the corresponding diagonal state [panel (b)], with $N \in \{2, 5, 10, 49\}$.

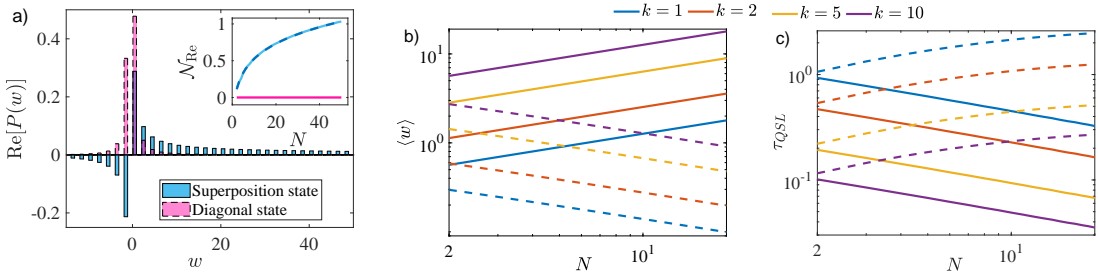

Figure 5: Distribution of the work done by the delta perturbation on the quantum system. Panel (a): Work MHQ distributions from initializing the system in the superposition state (3) (light blue) and in the corresponding diagonal state (pink), with $N = 50$. The inset shows the growth of the non-positivity functional $\mathcal{N}_{\mathrm{Re}}$ with $N$; the fit of the curve gives us $\mathcal{N}_{\mathrm{Re}} \approx 1.08(N^{0.17} - 1)$ (dashed line). Panels (b)-(c): Average work done by the perturbation and quantum speed limit $\tau_{\mathrm{QSL}}$, respectively, as a function of $N$, assuming the state of Eq. (3) (solid line) and the corresponding diagonal state (dashed lines) as initial states, for $k = 1, 2, 5, 10$.

be computed analytically. In particular, the overlap for $n = 0$ is given by [55]:

$$\Lambda_{m,0} = (-1)^{m/2} \sqrt{\frac{2^{m+1}}{(m+1)!} \frac{\Gamma\left(\frac{m+1}{2}\right)}{\pi}}, \tag{7}$$

where $\Gamma$ denotes the Gamma function. Then, for an arbitrary $n$, the overlap can be computed recursively, using the relation

$$\frac{\Lambda_{m,n}}{\Lambda_{m,n-2}} = -\sqrt{\frac{n-1}{n}} \frac{m-n+3}{m-n+1}. \tag{8}$$

Fig. 5 illustrates how the presence of quantum coherence in the initial state of the system leads to drastically different results after a sudden quench of its Hamiltonian. The real part of the KDQ — also known as Margenau-Hill quasiprobability (MHQ) [2, 16, 19, 56–58] — distribution of the work done by the defect, is shown in Fig. 5(a) and reveals non-positivity via negative probabilities. This amount of non-positivity can be quantified by [14] $\mathcal{N}_{\mathrm{Re}} \equiv -1 + \sum_{m,n} |\mathrm{Re}\, q_{m,n}|$, which exceeds zero whenever the system is initialized in a state with quantum coherence, and grows with a power of $N$ [inset of Fig. 5(a)].

In Fig. 5(a), we observe that negative work instances $\Delta E$ have negative probability to occur, meaning the work done by the defect is greater on average than when the system is initialized in a diagonal state. This emerges in the behaviour of the average work $\langle w \rangle$ done by the delta perturbation as a function of $N$ [see Fig. 5(b)]. For all four defect strengths $k$ considered, we observe two opposite behaviours of $\langle w \rangle$, depending on the initial state of the system. When the initial state is a coherent superposition, $\langle w \rangle$ grows with $N$, while it decreases in the absence of initial coherence. Initializing the system in the state (3), we can derive an analytical expression for $\langle w \rangle$ valid for large $N$ (see Appendix B for details):

$$\langle w \rangle = \frac{k}{\sqrt{2\pi}} \frac{1}{N} \left| 1 + \frac{\zeta(1/4)}{\pi^{1/4}} \right|^2 \cong k\sqrt{N}\, c, \tag{9}$$

where $c = \frac{8\sqrt{2}}{(9\pi)}$ and $\zeta$ is the Riemann zeta function.

Regarding the variance of work $\mathrm{Var}(w)$, we get:

$$\mathrm{Var}(w) = \langle \hat{V}^2 \rangle = \frac{k}{\sqrt{2\pi}} \langle w \rangle \left( 1 + \frac{\zeta(1/2)}{\sqrt{\pi}} \right), \tag{10}$$

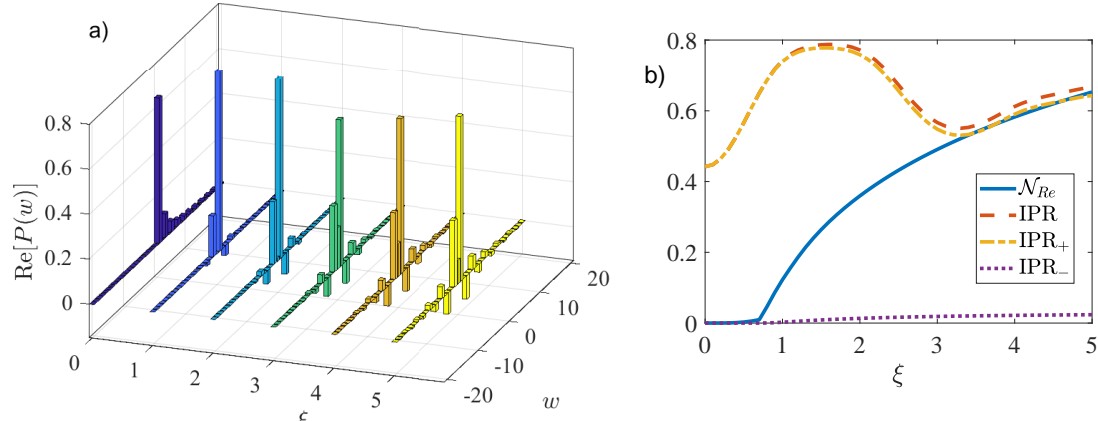

Figure 6: (a) MHQ distribution of the work done by a delta perturbation of strength $k = 100$ on a particle that is initialized in the coherent state Eq. (12), for $\xi \in \{0, 1, 2, 3, 4, 5\}$. This 3D bar plot illustrates how both the shape and support of the MHQ work distribution evolve as $\xi$ increases. (b) Comparison between the non-positivity functional $\mathcal{N}_{\mathrm{Re}}$ of the MHQ work distribution, and the inverse population ratio defined in Eq. (13). We also plot the two contributions of the inverse population ratio, IPR+ and IPR−, obtained by summing only over the positive and negative values of MHQs, respectively. The latter quantities are relevant in our analysis, as they quantify the localization or spread of negative (non-classical) work realizations across the whole work distribution.

as detailed in Appendix B. Even though the Riemann zeta function $\zeta(s)$ has a well-defined value at $s = 1/2$ through analytic continuation, the original Dirichlet series definition, which enters the formula for $\mathrm{Var}(w)$, does not converge at this point. Therefore, we may conclude that also $\mathrm{Var}(w)$ fails to converge.

Finally, the average work contributes to the orthogonalization velocity of the quantum system due to the sudden interaction with the defect. To characterize the process, we resort to the quantum speed limit $\tau_{\mathrm{QSL}}$ (defined in Ref. [59] and derived for our case-study in the Appendix C) that reads as

$$\tau_{\mathrm{QSL}} \equiv \frac{1 - |\nu(\tau)|}{|\langle w \rangle|}. \tag{11}$$

Fig. 5(c) shows that, when quantum coherence is included in the initial state of the system, $\tau_{\mathrm{QSL}}$ linearly decreases with $N$ for all values of $k$, which is a signature of a quantum speed-up of the orthogonalization time. An opposite trend is observed if quantum coherence is absent.

## 3.1   Single particle in an initial coherent state

Similar short-time behaviours of $|\nu(t)|$ can be observed [55] by initializing the quantum system in the coherent state

$$|\psi(0)\rangle = e^{-|\xi|^2/2} \sum_{n=0}^{N} \frac{(-1)^{n/2} \xi^n}{\sqrt{n!}} |n\rangle, \tag{12}$$

where $\xi \in \mathbb{C}$ denotes the coherent amplitude, and the additional factor $(-1)^{n/2}$ is introduced in analogy with the equal superposition state defined in Eq. (3).

In Fig. 6(a) we examine the MHQ distribution of the work $w$ done by the instantaneous $\delta$-like perturbation on the system, as a function of the coherent amplitude $\xi$. As $\xi$ increases, the work distribution displays rich features both in its shape and in its degree of asymmetry. In particular,

when $\xi = 0$ (blue histogram on the left of Fig. 6(a)), the initial state coincides with the ground state of the QHO that commutes with the unperturbed Hamiltonian. Hence, the distribution is always positive. Increasing $\xi$, the distribution progressively acquires negative regions. The associated non-positivity $\mathcal{N}_{\text{Re}}$ shown in panel (b), solid line, tends to zero for $\xi \to 0$, while it monotonically increases when $\xi$ grows.

From Fig. 6(a), we can observe that quasiprobabilities $q_{m,n}$ different from zero spread over the support of the work distribution. To quantify this spreading, as well as the corresponding localization properties, we introduce the following Inverse Population Ratio (IPR):

$$\text{IPR} = \sum_{m,n} \left( \text{Re}\, q_{m,n} \right)^2 . \tag{13}$$

The IPR of Eq. (13) is able to measure the effective spread of the work distribution, as attains its maximum value when the distribution comprises a single term (i.e., one probability is equal to one and all the other probabilities are vanishing) and decreases when the distribution delocalizes.

For this analysis, we distinguish between two contributions to the IPR (13) that sum to the IPR itself: IPR+ and IPR−, which are obtained similarly to Eq. (13) but by summing only over the positive and negative values of MHQs Re$\{q_{m,n}\}$, respectively. The quantity IPR$_-$ provides a measure of how the negative parts of the MHQ work distribution are spread across its support. It is worth observing that IPR$_-$ gives a different information about the negativity of the MHQ distribution, with respect to the non-positivity functional $\mathcal{N}_{\text{Re}}$. This is because $\mathcal{N}_{\text{Re}}$ is the cumulative sum of the modulus of each negative contribution of the distribution. Hence, an increase in $\mathcal{N}_{\text{Re}}$ indicates an enhancement of the non-classicality of the distribution, while a growing IPR$_-$ suggests that such negative values have a broad extension over the distribution.

Interestingly, in Fig. 6(b), both the IPR and IPR+ exhibit a pronounced bump at intermediate values of $\xi$. This feature arises because, in such a regime for $\xi$, the distribution tends to localize transitioning from being entirely positive (for small $\xi$) to becoming nearly symmetric around $w = 0$. The partial localization associated with this crossover enhances the IPR values, and this is associated with work realizations $w < 0$ that have a non-negligible probability. On the contrary, for larger $\xi$, the distribution spreads again symmetrically between positive and negative $w$, leading to a subsequent decrease of the IPR and IPR$_+$. Moreover, the non-positivity of the distribution continues to increase, reflecting at the same time both its broadening and a more balanced structure of their positive and negative components.

## 3.2   Two fermions

Let us consider again the case of two fermions initially prepared in the anti-symmetric state defined in Eq. (5), perturbed by a localized defect with interaction strength $k$. Compared with the single-particle case presented in Fig. 5, the MHQ distribution of the work performed by the defect exhibits a markedly larger loss of positivity. This enhancement of non-positivity can be attributed to the increased dimensionality of the Hilbert space, which enriches the structure of quantum coherences and interference effects. At the same time, in analogy with the single-particle scenario, the coherence present in the initial antisymmetric state induces a broader distribution of possible work values. The quench generated by the $\delta$-like potential activates multiple transitions across the two-particle energy manifold, giving rise to a more intricate and delocalized MHQ work profile.

Figure 7 compares the work MHQ distributions obtained initializing the system in the superposition antisymmetric state (light blue) or in the corresponding diagonal state (pink), respectively. In the former case the distribution exhibits a broader distribution, which extends towards both positive and negative values of $w$ that have been 'activated' by the $\delta$ perturbation. In contrast, the diagonal state produces a purely positive and more localized distribution, which is consistent with the absence of quantum interference effects.

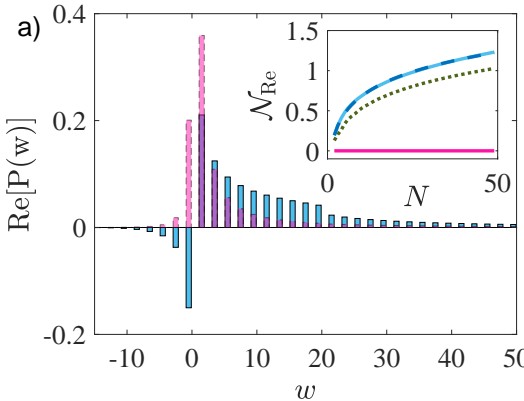

Figure 7: MHQ distribution of work for two fermions initialized in the anti-symmetric state (5) (light blue) and in the corresponding diagonal state (pink), with $N = 50$. The inset of the figure shows the growth of the non-positivity functional $\mathcal{N}_{\text{Re}}$ with $N$, following the fit $\mathcal{N}_{\text{Re}} \approx 3.83(N^{0.07} - 1)$ (dashed line); the green dotted line denotes the value of the non-positivity functional in the single-particle case.

In the inset of Fig. 7 we quantify the emergence of non-classical features in the MHQ distribution of work, through the plot of the non-positivity functional $\mathcal{N}_{\text{Re}}$ with respect to the number $N$ of initial superposition states. The data follow a slow algebraic growth given by $\mathcal{N}_{\text{Re}} \approx 3.83(N^{0.07}-1)$. The comparison with the single-particle reference (green dotted line) highlights how increasing the system size (here, following fermionic antisymmetrization) enhances the degree of negativity and, consequently, the non-classicality of the work statistics.

## 4 Experimental implementation

To experimentally probe the coherence-enhanced orthogonalization in one- or few-fermion systems, we consider a minimal setup based on ultracold atomic ensembles held in optical tweezers. Fermionic alkaline-earth-like atoms could be used to engineer both the system and the interacting localized defect, exploiting state-selective potentials [60, 61] and the tunable interactions between the $^1S_0$ and $^3P_0$ clock states [62–64]. Specifically, a single atom playing the role of the localized defect can be first initialized in the $^1S_0$ state and cooled to the motional ground state of an optical tweezer, near the $^1S_0$-state tune-out wavelength [61, 65]. Then, a system composed of a single or few $^1S_0$ atoms in a different nuclear spin state can be separately cooled to the lowest motional levels of a wider, yet highly anisotropic, harmonic trap. By dynamically acting on the trapping potential, possibly with optimal control sequences [66], a coherent state of atomic motion can be prepared along the weakest confinement direction [67, 68]. Exploiting sideband-driving with a clock transition, motional level superpositions can also be prepared [69]. The effective defect strength can be tuned by varying the spatial overlap of the two traps, the quench dynamics being initiated by a Ramsey clock $\pi/2$-pulse on the $^1S_0 \rightarrow {}^3P_0$ clock transition for the defect atom. Ramsey interferometry can be utilized to measure the time-dependent overlap and the system's coherence decay [30, 38, 41]. The impact of motional coherence on the dynamics for a chosen initial state could be quantitatively studied by letting the prepared superposition decohere over a variable amount of time into a statistical mixture before performing the interaction quench.

# 5   Conclusions

In this paper we have highlighted the profound influence of quantum coherence on the non-equilibrium dynamics developed by a quantum system coupled with a localized defect. This interaction is modelled by a quench transformation that leads to coherence-induced orthogonalization if the system initial state is in a coherent superposition of $N$ energy eigenstates. The LE modulus decays for large $N$ following a trend that resembles that of Anderson's OC. Similar short-time behaviours of $|\nu(t)|$ can be observed by initializing the quantum system in a coherent state, and even if the system is formed by two fermions. Despite differences resulting from the initial state and the number of fermions [55], in all the cases we analyzed, initial-state coherences not only modify the decay of the LE, but also enhance the average work done by the defect, i.e., the energy transfer to the system. This fact entails an evident quantum speed-up of the orthogonalization process.

Having obtained results for single and few fermions, a natural extension of this work is to explore how initial quantum coherence in a Fermi gas, for instance as induced by superpositions of few motional states, could modify the orthogonalization dynamics with respect to the established Anderson's scenario whereby the Fermi sea is initially in its equilibrium state. Furthermore, one could investigate how to exploit a delta perturbation with time dependent strength as a novel tool for quantum state engineering. To do that, optimal control [70] and reinforcement learning [71] routines could be considered. Considering the proposed experimental realization, the scenario of a not fully-localized impurity with significant quantum fluctuations remains to be explored, establishing connections with the nonequilibrium dynamics of Fermi polarons [30, 32, 72–74]. Finally, for quantum sensing purposes [75], one would determine — with theoretical arguments — what is the optimal state (possibly, a superposition one) of a quantum probe for estimating the strength of a localized defect interacting with it. Quantum dots may be the natural platform to realize such probes, as they are highly sensitive to charge fluctuations in their local environment. This sensitivity allows them to infer their distance to a defect with nanometer-scale resolution [76, 77].

## Acknowledgements

B.D., F.S. and S.G. acknowledge financial support from the PNRR MUR project PE0000023-NQSTI funded by the European Union—Next Generation EU. G.D.C. and S.G. acknowledge support from the Royal Society Project IES\R3\223086 "Dissipation-based quantum inference for out-of-equilibrium quantum many-body systems". F.S. acknowledges financial support also from the European Union under the Horizon 2020 research and innovation programme (project OrbiDynaMIQs, GA No. 949438) and under the Horizon Europe program HORIZON-CL4-2022-QUANTUM-02-SGA (project PASQuanS2.1, GA no. 101113690).

# A   Cusp singularities of the Loschmidt echo

In the limit of strong perturbation $k \to \infty$, the overlaps $\Lambda_{m,n}$ [see Eq. (8)] can be computed analytically. This leads to closed-form expression of the LE, including its periodicity and cusps. Here, for simplicity of calculation, we consider the following superposition state (parametrized by the angles $\theta$ and $\phi$) as initial state: $|\psi(0)\rangle = \cos\left(\frac{\theta}{2}\right)|0\rangle + e^{i\phi}\sin\left(\frac{\theta}{2}\right)|2\rangle$. Starting from this initial state

and using the expression of KDQs as a function of the overlaps $\Lambda_{m,n}$, the LE is

$$
\begin{aligned}
\nu(t) &= \sum_{m,n} e^{-i\Delta E_{m,n}} q_{n,m} = \sum_{m\,even} \left( e^{-i(m+1)t} q_{0,m} + e^{-i(m-1)t} q_{2,m} \right) = \\
&= \sum_{m\,even} \left( e^{-i(m+1)t} \left( \frac{1+\cos\theta}{2} \Lambda_{m,0}^2 + e^{i\phi} \frac{\sin\theta}{2} \Lambda_{m,0}\Lambda_{m,2} \right) \right. \\
&\qquad\qquad \left. + e^{-i(m-1)t} \left( e^{-i\phi} \frac{\sin\theta}{2} \Lambda_{m,0}\Lambda_{m,2} + \frac{1-\cos\theta}{2} \Lambda_{m,2}^2 \right) \right) = \\
&= \sum_{n=0}^{+\infty} e^{-i2nt} \frac{\Lambda_{2n,0}^2}{2} \left( e^{-it}(1+\cos\theta) + 2\cos(\phi-t) \times \frac{\sin\theta}{\sqrt{2}} \frac{2n+1}{2n-1} + e^{it} \frac{1-\cos\theta}{2} \left( \frac{2n+1}{2n-1} \right)^2 \right),
\end{aligned}
\tag{A.1}
$$

where we have used both the recursive relation of Eq. (8) in the main text to express $\Lambda_{m,2}$ as a function of $\Lambda_{m,0}$, and the energies $E'_m = m + 1 + 1/2$ and $E_n = n + 1/2$. The square of the overlap $\Lambda_{2n,0}$ can be rewritten as $\Lambda_{2n,0}^2 = \frac{2}{\pi} \binom{2n}{n} \frac{1}{2^{2n}(2n+1)}$. Hence, using the Taylor expansion

$$
\arcsin z = \sum_{k=0}^{+\infty} \binom{2k}{k} \frac{z^{2k+1}}{2^{2k}(2k+1)} \quad \text{with} \quad |z| \le 1,
\tag{A.2}
$$

the LE simplifies as

$$
\nu(t) = \frac{2}{\pi} \left[ \arcsin(e^{-it}) - \frac{\sqrt{1-e^{-i2t}}}{4} \left( e^{it}(\cos\theta - 1) + 2\sqrt{2}\cos(\phi - t)\sin\theta \right) \right].
\tag{A.3}
$$

It is now evident that for $t = s\pi$, with $s \in \mathbb{N}$, the modulus of LE is simply

$$
|\nu(t = s\pi)| = \frac{2}{\pi} \left| \arcsin e^{is\pi} \right| = \frac{2}{\pi} \left| (-1)^s \frac{\pi}{2} \right| = \left| e^{is\pi} \right| = 1 \quad \forall \theta, \phi.
\tag{A.4}
$$

Moreover, the time-derivative of $|\nu(t)|$ diverges in $t = s\pi$, which is the signature of the observed cusp behaviour.

## B   Average and variance of work KDQ distribution

Let us consider a generic initial superposition state $|\psi(0)\rangle = \sum_n \alpha_n |\psi_n\rangle$. Thus, the analytical expression of the average work done by the delta perturbation can be calculated analytically as follows:

$$
\langle w \rangle = \langle \psi(0)| \hat{V} |\psi(0)\rangle = \sum_n \alpha_n^* \sum_{n'} \alpha_{n'} \int dx\, \psi_n(x)^* k\,\delta(x) \psi_{n'}(x) =
$$

$$
= k \sum_{n,n'} \alpha_n^* \alpha_{n'} \psi_n(x=0)^* \psi_{n'}(x=0) = \frac{k}{\sqrt{2\pi}} \left| \sum_{n\,even} (-1)^{n/2} \alpha_n c_n \right|^2,
\tag{B.1}
$$

where we have used the equality $\psi_n(x=0) = (-1)^{n/2}\sqrt{2^n}(n-1)!!/((2\pi)^{1/4}\sqrt{2^n n!})$ and we have defined the quantity $c_n \equiv (n-1)!!/\sqrt{n!}$. In first place, we can conclude that the average work is always positive.

Now, let us specifically calculate $\langle w \rangle$ for the initial state of Eq. (3) in the main text. For such a case, the summation inside the square modulus of Eq. (B.1) becomes:

$$
\sum_{n\,even} (-1)^{n/2} \alpha_n c_n = \sum_{n\,even} (-1)^{n/2} \frac{(-1)^{n/2}}{\sqrt{N}} \frac{(n-1)!!}{\sqrt{n!}} = \frac{1}{\sqrt{N}} \sum_{s=0}^{N/2} \frac{(2s-1)!!}{\sqrt{(2s)!}} = 1 + \frac{1}{\sqrt{N}} \sum_{s=1}^{N/2} \frac{\sqrt{(2s)!}}{2^s s!}.
\tag{B.2}
$$

Then, using the Stirling approximation $n! \cong \sqrt{2\pi n}(n/e)^n$ for large $N$, Eq. (B.2) simplifies to

$$1 + \pi^{-1/4} \sum_{s=1}^{N/2} s^{-1/4} = 1 + \frac{\zeta(1/4)}{\pi^{1/4}} , \tag{B.3}$$

where we have substituted the Riemann zeta function, defined as $\zeta(s) \equiv \sum_{s=1}^{\infty} \frac{1}{n^s}$. Thus, substituting Eqs. (B.2)-(B.3) in Eq. (B.1), the average work is

$$\langle w \rangle = \frac{k}{\sqrt{2\pi}} \frac{1}{N} \left| 1 + \frac{\zeta(1/4)}{\pi^{1/4}} \right|^2 . \tag{B.4}$$

Albeit Eq. (B.4) is a closed-form expression for the average work $\langle w \rangle$, it is worth noting that the Riemann zeta function is well defined for $s > 1$, while for $0 < s < 1$ one has to use its analytical extension in the complex plane that however can lead to unphysical results. Therefore, we take a step back and approximate the series in Eq. (B.3) with an integral:

$$\sum_{s=1}^{N/2} s^{-1/4} \approx \int_1^{N/2} s^{-1/4} \, ds = \frac{4}{3} s^{3/4} \Big|_1^{N/2} . \tag{B.5}$$

The approximation is more accurate for increasing $N$. In this way, substituting Eq. (B.5) in Eq. (B.3) and then in Eq. (B.1), the average work reads as

$$\langle w \rangle = \frac{k}{\sqrt{2\pi}} \frac{1}{N} \left| 1 + \pi^{-1/4} \frac{4}{3} \left( \left( \frac{N}{2} \right)^{3/4} - 1 \right) \right|^2 \approx \frac{k}{\sqrt{2\pi}} \frac{1}{N} \left| \pi^{-1/4} \frac{4}{3} \left( \frac{N}{2} \right)^{3/4} \right|^2 = k\sqrt{N} \frac{8\sqrt{2}}{9\pi} . \tag{B.6}$$

Eq. (B.6) provides us an alternative expression of the average work that we have verified numerically. From Eq. (B.6), we can observe a linear dependence of $\langle w \rangle$ on the intensity $k$ of the delta perturbation and a square-root dependence on $N$, number of the unperturbed Hamiltonian's eigenstates.

The computation of the variance of work $\mathrm{Var}(w)$ follows the same methodology we used for the average work. For an initial superposition state with real coefficients $\alpha_n$, the variance of work $\mathrm{Var}(w) = \langle \hat{H}\hat{V} \rangle - \langle \hat{V}\hat{H} \rangle + \langle \hat{V}^2 \rangle$ is equal to the second statistical moment of the work distribution $\langle w^2 \rangle$ and reads as

$$\mathrm{Var}(w) = \langle \hat{V}^2 \rangle = \frac{k}{\sqrt{2\pi}} \langle w \rangle \left( 1 + \frac{\zeta(1/2)}{\sqrt{\pi}} \right) . \tag{B.7}$$

From Eq. (B.7), we can conclude that $\mathrm{Var}(w)$ diverges due to the behaviour of the Riemann zeta function in $1/2$, which does not converge in the standard sense. Indeed, the analytical extension of $\mathrm{Var}(w)$ gives the value $\zeta(1/2) \approx -1.46$, implying an unphysical negative variance of the work distribution.

## C Quantum speed limit

In this section, we show that the expression of the quantum speed limit in Ref. [59] is valid for a generic initial state and non-commuting observables.

For this purpose, we start defining the Bures angle derived from the LE: $\mathcal{L}(t) \equiv \arccos |v(t)|$. Then, we look for an upper bound of the Bures angle $\mathcal{L}(t)$ by computing its time derivative and taking the absolute value of the resulting expression:

$$\partial_t \mathcal{L}(t) \leq \left| \partial_t \mathcal{L}(t) \right| = \left| \frac{\partial_t v(t)}{\sqrt{1 - |v(t)|^2}} \right| . \tag{C.1}$$

Rearranging the terms in Eq. (C.1), we have that

$$\dot{\mathcal{L}} \sin \mathcal{L} \le \left| \partial_t \nu(t) \right|. \tag{C.2}$$

Integrating the left-hand-side of Eq. (C.2) over time yields:

$$\int_0^\tau dt \dot{\mathcal{L}} \sin \mathcal{L} = \cos \mathcal{L}(0) - \cos \mathcal{L}(\tau) = 1 - \left| \nu(\tau) \right|, \tag{C.3}$$

while for the corresponding right-hand-side we have that

$$\int_0^\tau dt \left| \partial_t \nu(t) \right| = \int_0^\tau dt \left| \sum_{n,m} (E_m' - E_n) e^{-i(E_m' - E_n)t} q_{n,m} \right| = \tau \left| \langle w \rangle \right|. \tag{C.4}$$

In conclusion, substituting Eq. (C.3) and Eq. (C.4) inside Eq. (C.2), we obtain the quantum speed limit bound of Ref. [59] that we have used in the main text:

$$\tau \ge \tau_{\text{QSL}} \equiv \frac{1 - |\nu(\tau)|}{|\langle w \rangle|}. \tag{C.5}$$

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
