# Peer review of "Orthogonalization speed-up from quantum coherence after a sudden quench"

_SciPost Physics_

## Round 1 · Referee Report · Anonymous (Referee 1) · 2025-12-17

Strengths

1- investigates the role of coherence on the decay of the Loschmidt echo. 2- initial decay rates shown to follow scaling law. 3- emergence of non-classicality shown in the spectral function.

Weaknesses

1- numerics seem not to be converged. 2- concrete comparisons not made to the original Anderson problem. 3- uncertainty about the impact of the main result. 4- very complicated experimental implementation.

Report

The authors investigate the speed up of orthogonalization due to the presence of coherences in the initial state. They focus on superpositions of single and two particle states, and compare with initial diagonal ensembles whose Loschmidt Echo decays slower due to a lack of initial coherence. The non-classicality of the initial states also shows up in the real part of the spectral function as negative probabilities, which contrasts to the diagonal state only having positive probabilities. The speed up in decay of the initial state is related to the orthogonality catastrophe in Fermi gases and an experimental implementation is proposed.

I find the work somewhat interesting, but I am unsure how truly non-trivial the results are. One problem I have is that the main results of the work are compared to Andersons OC in Fermi gases, but there is not a lengthy discussion or comparison between the two phenomena. On first look the superposition state with coherence seems very similar to a Fermi gas in which the same single particle states are taken in the Slater determinant. In such a case the dynamics of the Fermi gas can be similarly described in terms of single particle overlaps as shown at the beginning of Section 3. Is there a fundamental difference between these two cases? Can the main result of this paper about the role of coherences also be applied to the Fermi gas? The second issue I have is with the results shown in the figures and their convergence. I have detailed my concerns below.

I have too many unresolved questions that need to be addressed before any decision on publication can be made.

Major comments: 1) I must comment on convergence issues that I notice in the results and also the discussions. When considering the dynamics of the LE, by construction it must equal 1 at t=0. This is not true in Fig.1(a), Fig.2(a), Fig.3(a), Fig.4(a,b).

The reason for this is due to not enough perturbed eigenstates taken in the single particle overlaps Λm,n. For each state n taken in the superposition the sum of |Λm,n|^2 over m should tend to 1 assuming m is sufficiently large. For small quenches in k the required number of perturbed eigenstates is small, but for the large quenches that you focus on, i.e. k=100, a large number of states are needed (100s to 1000s).

I recommend that this issue is fixed by checking the sum of the norm of the overlaps is sufficiently close to 1 (lets say realistically >0.999). If this cannot be reached by increasing the number of perturbed eigenstates I suggest focussing on smaller quenches, k=50 or k=20.

2) This convergence issue is particularly relevant when doing an infinite quench, k->\infty, as discussed in Sec.3. For the same reason as above it will be impossible for any sum to converge as an infinite repulsive barrier forces the perturbed wavefunctions to vanish at x=0. The initial state (without the barrier) will never be able to be entirely projected on the infinite barrier basis as at x=0 the correct density will never be resolved. Furthermore, the average energy post-quench is divergent, which leads to your unphysical calculation of the variance of work. This is all discussed in this paper [Kehrberger et al. Phys. Rev. A 97, 013606 (2018)] in Section III. On the other hand, in the quench from infinity to zero the energy converges, so you do not have these issues.

I recommend that the discussion around the infinite quench and its divergence is discussed in more detail, along with the reference.

3) "As N increases, the system remains at the minimum of |ν(t)| for a longer duration if quantum coherence are included in the initial state, as shown in panel (a), thus highlighting a more stable orthogonalization process. "

Maybe a nitpick, but for your choice of superposition states orthogonality is never reached, whereas for the diagonal state orthgonality is definitely reached for N=100 (also very close for N=10). While the decay of the LE can be sped up, true orthogonality is not reached faster.

In fact, I wonder if orthogonality can ever be reached for the superposition state? For example, at t=pi/2 can you find an expression for the LE and how it scales with N?

4) Could the constant LE value reached for the superposition state be related to the initial coherences preventing the states becoming orthogonal? Can the degree of coherence be quantified (by summing all the off diagonal terms), and if so does it decay after the quench or does it saturate when the LE does?

5) At the end of section 2.1 there is a comment made about DPTs in this system: "It is worth noting that, differently to the canonical manifestation of DPT phenomenology in many-body system, the single-particle case lacks a spatial order parameter that prevents the revival of |ν(t)| to 1."

This I do not follow, particularly with respect to a complete revival of the initial state |ν(t)| = 1. Perfect revivals of the initial state can occur in many different single particle systems, for instance quenches of the trap frequency in the harmonic oscillator, or Talbot oscillations in box potentials. In the case of the delta function perturbation in the harmonic oscillator, perfect revivals can also occur for the aforementioned quenches from infinity to zero as the final Hamiltonian spectrum is harmonic. Conversely, quenching on a finite delta function breaks the harmonicity of the spectrum and therefore complete revivals will no longer occur as each perturbed eigenstate oscillates slightly out of phase, as in your case.

Also, with respect to DPTs, as far as I understand the cusp singularity only occurs at orthogonality, |ν(t)| -> 0, opposite to what is discussed here. I would suggest removing any reference to DPTs.

6) in Section 3.1 when the coherent state is introduced the additional factor (-1)^{n/2} is commented on, namely to match Eq.3. I am curious as to why this is needed in the first place? Would you get totally different results if the coherence terms were all positive? Whatever the reason it should be justified in the text.

7) At the end of Section 2 it is commented that your results with the superposition state is "somewhat reminiscent" of the Anderson's OC with a many-body Fermi sea. But isn't it exactly the same? For instance, the Fermi sea also contains coherences, which according to your analysis are responsible for accelerating the decay of the LE. If you start with an initial N-particle Fermi gas which is in an excited state where only the lowest N even states are occupied, while the odd states are unoccupied, isn't this very similar (or exactly the same just in a different basis?) to your superposition state defined by Eq.3? If so, is this the answer to my 6th question and the choice of (-1)^{n/2} ?

8) Again, related to the last 2 comments, when discussing the MHQ distribution and the growth of non-positive probabilities, would the N-body Fermi gas (ground state or excited states) also have negative probabilities?

9) In the abstract and introduction there is mention of "A direct manifestation of this effect is the discrete counterpart of an infinite discontinuity in the quasiprobability distribution of work done by the defect", but this is never explained or elaborated upon in the discussions.

10) Is the experimental proposal realistic? Creating a superposition over two levels seems possible, but over 49?

Minor Comments: 11) When discussing Fig.1 you say "Interestingly, γ tends to become constant for k large (in Fig. 1, γ ≈ 0.41 for k = 100)." - I would append this to note that this occurs at long times.

12) Should the sum in Eq.5 also be over only even states?

13) Immediately after Eq.5 where you have the scaling law should the 'M' be 'N'? (Also superposition size 'M' is mentioned a couple of times in this paragraph).

14) In Section 3.2 when discussing the two fermion results Fig.7 should be referenced in the first paragraph when discussing those results.

15) The number of states N of the superposition state is not listed for the MHQ distributions in any of the figures.

Requested changes

Major comments and minor comments addressed.

Recommendation

Ask for major revision

---

## Editorial Decision

in_refereeing